

# The PERSIANN Family of Global Satellite Precipitation Data: A Review and Evaluation of Products

5 Phu Nguyen[1], Mohammed Ombadi[1], Soroosh Sorooshian[1], Kuolin Hsu[1], Amir AghaKouchak[1], Dan Braithwaite[1], Hamed Ashouri[1], Andrea Rose Thorstensen[1]

10 [1]Center for Hydrometeorology and Remote Sensing, Department of Civil and Environmental Engineering, University of California Irvine, Irvine, CA, USA

*Correspondence to*: Phu Nguyen (ndphu@uci.edu)



## Abstract

Over the past two decades, Precipitation Estimation from Remotely Sensed Information using Artificial Neural Networks (PERSIANN) products have been incorporated in a wide range of studies. Currently, PERSIANN offers several precipitation products based on different algorithms available at various spatial and temporal scales, namely, PERSIANN, PERSIANN-CCS and PERSIANN-CDR. The goal of this article is to first provide an overview of the available PERSIANN precipitation retrieval algorithms and their differences. Secondly, we offer an evaluation of the available operational products over the Contiguous United States at different spatial and temporal scales using Climate Prediction Center (CPC) Unified gauge-based analysis as a benchmark. Finally, the available products are intercompared at a quasi-global scale. Furthermore, we highlight strength and limitations of the PERSIANN products and briefly discuss the expected future developments.

# 1 Introduction

## 1.1 Precipitation and Satellite-based observation

Precipitation is an integral part of the Earth's hydrologic cycle, playing a foremost role in its water and energy balance. Accurate, uninterrupted and uniform observation of precipitation represent important inputs for research and operational applications. The resilience and capacity of societies to react and adapt to climate extremes such as storms, floods and droughts are greatly enhanced with a long term historical record of precipitation. Practical applications include use of precipitation Intensity-Duration-Frequency (IDF) information for infrastructure design and use of near real-time precipitation data in development of early warning systems and disaster management planning. Moreover, the observation of precipitation is essential for understanding Earth's climate, its underlying variabilities and trends. In turn, climatic understanding can improve our ability to forecast extreme events and enables informative strategic planning and decision making on issues related to water supply, both in quantity and quality.

Precipitation measurement continues to represent a great challenge for the scientific community mainly due to its spatiotemporal variations in intensity and duration (Sorooshian et al., 2011). The three primary instruments used for measurement of precipitation are gauges, radar and satellite. Rain gauges provides direct measurement of precipitation; however, it suffers from intermittent coverage over most continents. Radar technology is not available in many countries and even in places where the technology is available, radar blockage by mountains is a major challenge. Both rain gauges and radar do not provide measurements over oceans. On the other hand, satellite-based precipitation measurements seem to be the most promising method to accurately observe precipitation over both land and ocean.



Satellite-based precipitation estimation techniques are mainly based on information from Geostationary Earth Orbiting (GEO) satellites and/or Low Earth Orbiting (LEO) satellites. GEO satellites are capable of providing images every 5-30 minutes in multiple spectral bands, but their spectral coverage is limited to visible and infrared wavelengths. On the other hand, LEO satellites are able to provide passive microwave (PMW) information about the hydrometeors directly relevant to surface

precipitation rates. Early techniques for developing satellite-based estimates of rainfall are briefly described in Hsu et al. (1997) which includes the analysis of individual pixel information such as (Arkin & Meisner 1987) and the analysis of cloud image types and their variations in time (Scofield et al., 1987). In 1997, the Tropical Rainfall Measurement Mission (TRMM) was launched. It carried the first orbital rainfall radar, which was used to calibrate passive microwave sensors on other satellites, resulting in significant improvements to rainfall retrievals over the tropical regions of the globe (Kummerow et al., 1998;

Kummerow et al., 2000; Simpson et al., 1988). The follow-up to the TRMM mission is the Global Precipitation Measurement (GPM). The GPM mission deploys an enhanced dual-frequency radar sensor. The GPM program aims to combine observations from multiple passive microwave sensors mounted on both pre-existing and newly-deployed satellites. The GPM satellite constellation has global coverage in the range (68°S - 68°N) with a return interval of 3 hours (Hou et al., 2014).

Today, several agencies and institutes provide satellite-based datasets which have been derived using different algorithms such

as the Climate Prediction Center (CPC) morphing technique CMORPH (Joyce et al., 2004), TRMM Multi-Satellite Precipitation Analysis TMPA (Huffman et al. 2007), NASA Integrated Multi-satellitE Retrievals for GPM (IMERG) (Huffman et al., 2015), NRL-Blend satellite rainfall estimates from the Naval Research Laboratory (NRL) (Turk et al., 2010) and Precipitation Estimation from Remotely Sensed Information using Artificial Neural Networks (PERSIANN; Hsu et al., 1997, 1999; Sorooshian et al., 2000). Both PERSIANN and CMORPH algorithms employ GEO and LEO satellite information to

estimate rainfall. However, while CMORPH uses GEO-IR cloud motion vectors to linearly interpolate PMW rainfall estimates between sensor overpasses (Joyce et al., 2004), PERSIANN algorithm determines a relationship between GEO infrared imagery and precipitation and adjusts the estimates using PMW rainfall estimates form LEO satellites.

## 1.2 PERSIANN products

Over the last two decades, the PERSIANN system of precipitation products have been developed at the Center for

Hydrometeorology and Remote Sensing (CHRS) at the University of California, Irvine in collaboration with NASA, NOAA and the UNESCO programme for the Global Network on Water and Development Information for Arid Lands (G-WADI) program. The PERSIANN family includes three satellite-based precipitation estimation products namely PERSIANN, PERSIANN-CCS and PERSIANN-CDR. The products are accessible through several web-based interfaces to serve the needs of researchers, professionals and general public including: CHRS iRain (http://irain.eng.uci.edu), Data Portal

(http://chrsdata.eng.uci.edu) and RainSphere (http://rainsphere.eng.uci.edu). These web-based interfaces provide different visualization, analysis and download capabilities.

PERSIANN products have also been used frequently for different studies by researchers in the fields of hydrology, water resources management and climate. These include evaluation of PERSIANN products against ground observations, other





satellite-based products and model simulations (Sorooshian et al., 2002; Yilmaz et al., 2005; Li et al., 2003; Miao et al., 2015; Nguyen et al., 2017; Mehran et al., 2014), application of PERSIANN products for modeling soil moisture (Juglea et al., 2010), prediction of runoff (Behrangi et al., 2011; Ashouri et al., 2016; Liu et al., 2017; AghaKouchak et al., 2010; Hsu et al., 2013), rainfall frequency analysis (Gado et al., 2017), tracking typhoons (Nguyen et al., 2014), monitoring drought (Katiraie-

Boroujerdy et al., 2016; AghaKouchak & Nakhjiri, 2013), assimilation into climate models (Yi, 2002), precipitation forecasting (Zahraei et al., 2013), and trend analysis (Nguyen et al., 2016; Damberg et al., 2014).

The main objective of this paper is to provide a concise and clear summary of the similarities and differences between the three products in terms of attributes and algorithm structure. Moreover, the paper aims to provide an evaluation of the performance of the products over the Contiguous United States (CONUS) using the Climate Prediction Center (CPC)

precipitation dataset as a baseline of comparison. Also, an assessment of the behaviour of PERSIANN family products over the globe (60°S - 60°N) is performed. The subsequent sections of this paper are organized as follows: Sect. 2 presents a brief description and comparison of the algorithm and attributes of each product. Sect. 3 and Sect. 4 provide the results of evaluation of the PERSIANN products over the CONUS and the globe respectively. In Sect. 5, conclusions are provided to pinpoint future development areas.

## 2. Attributes and algorithms structures of PERSIANN products

In this section, general descriptions of the attributes and algorithm structures of the PERSIANN products are provided and compared. A comparison of the attributes of the three products is shown in Table 1. The percentage of missing data is shown over the period (2003-2015) and its spatial distribution can be seen in Fig. 1 for each of the products. It should be noted that

missing data will be due to input geostationary satellite data unavailability.

### 2.1 Precipitation Estimation from Remotely Sensed Information Using Artificial Neural Networks (PERSIANN)

The PERSIANN algorithm, developed in 1997, is based on the synergy between the sparsely sampled information from low Earth orbiting (LEO) satellites and the high frequency samples from geostationary (GEO) satellites. With regard to GEO

imagery, PERSIANN originally used longwave infrared imagery as the primary input to the algorithm; however, it was later extended to include daytime visible imagery as well. The passive microwave imagery from low earth orbiting (LEO) satellites is used to continuously adapt the parameters of the model.

The PERSIANN algorithm is an artificial neural network (ANN) model based on a multilayer neural feedforward network (MFN) known as the Modified Counter Propagation Network (Hsu, 1996). This hybrid model consists of two processes: firstly,

the infrared (10.2-11.2 μm) images are transformed into the hidden layer through an automatic clustering process to form what is known as a self-organizing feature map (SOFM). The purpose of this process is to detect and classify patterns in the input



data. Then, the discrete SOFM clusters in the hidden layer are mapped to the continuous space of outputs (i.e. rainfall rate). Both processes of input-hidden and hidden-output transformations involve parameter estimation which is routinely performed by incorporating Passive Microwave (PMW) rainfall from Low Earth Orbiting satellites. It should be noted that parameter estimation in each process can be performed separately by training the model for the former while a supervised learning strategy is used for the latter. PERSIANN data is available for public use through the CHRS Data Portal http://chrsdata.eng.uci.edu. For a comprehensive description of the original PERSIANN algorithm and the several enhancements added to it, interested readers should refer to (Hsu et al., 1997, 1999, 2007), (Sorooshian et al., 2000, 2002).

## 2.2 PERSIANN-Cloud Classification System (PERSIANN-CCS)

PERSIANN-CCS, (Hong et al., 2004), is an example of cloud-patch-based algorithms where the features are extracted from the cloud coverage under specified temperature thresholds. Algorithms with a similar concept were developed earlier such as the Griffith-Woodley technique (Griffith et al., 1978) (Woodley et al., 1980), the Convective-Stratiform technique (Adler & Negri, 1988) and the method proposed by Xu (Xu et al., 1999). The PERSIANN-CCS algorithm utilizes more information from the infrared cloud images compared to PERSIANN by performing segmentation of the cloud image under different temperature thresholds. The algorithm involves four steps: firstly, the infrared cloud image is segmented based on different temperature thresholds using an incremental temperature threshold (ITT) approach. Next, features such as coldness, geometry and texture are extracted from the segmented images in an attempt to distinguish between different cloud types and the cloud patch is identified as one of 400 classified cloud types. In the third step, the SOFM clustering algorithm described earlier in the PERSIANN algorithm is used to classify the cloud extracted features into distinct categories. Finally, for each feature group resulting from the previous step, a relationship between brightness temperature and rainfall rate is developed using histogram matching and nonlinear exponential function fitting (Hong et al., 2004). A PMW rainfall calibrated PERSIANN-CCS algorithm was developed beginning in 2014 and has been implemented as part of the NASA GPM IMERG algorithm (Karbalaee et al., 2017). At this time, the PMW calibrated PERSIANN-CCS products are available after 2014. PERSIANN-CCS data is available for public use through the CHRS Data Portal http://chrsdata.eng.uci.edu. For a comprehensive description of PERSIANN-CCS and PMW calibrated algorithms, interested readers should refer to (Hong et al., 2004) and (Karbalaee et al., 2017).

## 2.3 PERSIANN-Climate Data Record (PERSIANN-CDR)

PERSIANN-CDR uses a modified PERSIANN algorithm in order to produce a historical record of precipitation estimates dating back to 1983. As mentioned earlier, the PERSIANN algorithm relies primarily on infrared imagery from GEO satellites as an input to the ANN model. Similarly, PERSIANN-CDR uses infrared imagery data from different international GEO satellites which is available starting from 1979 at 10 km spatial resolution and 3hr temporal resolution (Rossow & Schiffer, 1991; Rossow & Garder, 1993; Knapp, 2008) and maintained by NOAA under the International Satellite Cloud Climatological



Project (ISCCP). However, unlike the PERSIANN algorithm where Passive Microwave imagery is used to update the parameters of the network, PERSIANN-CDR alternatively uses the National Centers for Environmental Prediction (NCEP) stage IV hourly precipitation to train the ANN model. Then, the algorithm is run with fixed parameters to estimate the historical data. An additional processing step is performed to reduce the bias in the PERSIANN-CDR estimates by incorporating Global

5    Precipitation Climatology Project (GPCP) monthly 2.5° precipitation data. The resulting PERSIANN-CDR estimates maintain monthly total precipitation consistent with the GPCP data (Ashouri et al., 2015). PERSIANN-CDR data is available for public use through the NOAA National Centers for Environmental Information (NCEI) at https://www.ncdc.noaa.gov/cdr/atmospheric/precipitation-persiann-cdr and the CHRS Data Portal http://chrsdata.eng.uci.edu. For a comprehensive description of the PERSIANN-CDR algorithm, interested readers should refer to (Ashouri et al., 2015).

## 3. Evaluation of PERSIANN products over the CONUS

The three products PERSIANN, PERSIANN-CCS and PERSIANN-CDR have been evaluated over the CONUS for the period starting from 2003 to 2015. The main dataset used as a reference for evaluation is NOAA Climate Prediction Center (CPC) Unified Gauge-Based Analysis of Daily Precipitation over CONUS (retrieved from ftp://ftp.cdc.noaa.gov/datasets),

hereafter will be referred to as CPC. CPC data has a spatial resolution of (0.25° x 0.25°), it was used as a baseline for evaluation in this study because it combines all ground-based information sources.

Figure 2 shows the average annual precipitation during (2003-2015) for the four datasets. The average annual precipitation of PERSIANN, PERSIANN-CCS, PERSIANN-CDR and CPC over the CONUS in mm is 793, 979, 916 and 852 respectively.

It can be clearly seen in Fig. 2 that PERSIANN-CDR better mirrors the precipitation patterns observed in the CPC data. This

is not surprising since PERSIANN-CDR, as detailed in Table 2, is a bias adjusted product utilizing the GPCP data. Over the Gulf States, PERSIANN and PERSIANN-CCS both tend to underestimate the mean annual precipitation, whereas PERSIANN-CDR has captured those patterns. The average annual precipitation of PERSIANN generally matches the pattern of PERSIANN-CDR and CPC although an underestimation over the northwestern United States (Washington, Oregon and North California) and a spurious overestimation in midwestern states are observed. With regard to PERSIANN-CCS, the

pattern and values of average annual rainfall differ considerably from the other products.

The above results represent comparison over a large time scale, as a result, errors in short time scale may cancel each other which may obscure important information. Therefore, a comparison based of daily precipitation data is needed to obtain a more insightful view about the data sets. In order to make PERSIANN data consistent with CPC in a daily scale, PERSIANN

data have been accumulated from 12z to 12z instead of the default (0-24) accumulation. This daily comparison is shown with both continuous metrics (correlation coefficient (CORR), root mean squared error (RMSE) and bias) in Fig. 3 and categorical metrics (probability of detection (POD) and false alarm ratio (FAR)) in Fig. 4. A general conclusion to be inferred from Fig.





3 is the disparity of correlation coefficient with relatively low values over the western states and high values extending from Gulf States in the south toward the northern part of the country for the three products. On the contrary, it can be seen that RMSE has better values over the western US and poor values over Gulf States and the eastern US. It is important to state that these two results don't provide a diagnosis about the products performance but rather can be attributed to differences in climatic zones. The western states on average receive less precipitation compared to Gulf and eastern states, and, RMSE will often have higher values where precipitation is higher. Additionally, results over the western US might have also been affected by lack of sufficient gauge data used in the interpolation process of creating the CPC gridded product. As for RMSE and bias, PERSIANN-CDR shows the best performance among all the PERSIANN products. In particular, the bias is almost zero across the country (except in some areas in Nevada, Washington, and parts of the central valley in California). As shown in Fig. 2, on average heavier rainfall events occur over Gulf states and the states along the east coast. With that in mind and based on the statistics shown in Fig. 3, one can see that PERSIANN-CDR captures such patterns with high correlation coefficient and very low bias. See Table 3 for a summary of the continuous metrics over the CONUS.

On the other hand, the categorical indices shown in Fig. 4 illustrate that the three PERSIANN products, PERSIANN, PERSIANN-CCS and PERSIANN-CDR have similar performance in terms of FAR with average values over the CONUS equivalent to 0.22, 0.29 and 0.29 respectively. However, PERSIANN has considerably lower probability of detection (POD) with an average value of 0.8 compared to 0.9 for both PERSIANN-CCS and PERSIANN-CDR. See Table 3 for a summary of the categorical metrics over the CONUS. It is noted that categorical indices do not distinguish between light and heavy rains. An alternative approach would be considering volumetric indices for evaluation of satellite products including Volumetric Hit Index (VHI) and Volumetric False Alarm (VFA) (AghaKouchak and Mehran, 2013). These indicators are based on volume of captured rain rather than counts. Previous studies show that while PODs of PERSIANN products might be rather low in some regions, the VHI is much higher indicating that PERSIANN products capture most of the volume of rain in reference products (see AghaKouchak and Mehran, 2013).

To further investigate the performance of the PERSIANN products, we narrowed down our analysis to extreme precipitation events. We looked at three classes of extreme indices, namely absolute threshold, percentile, and maximum indices; these indices are defined in Table 4. Results for SDII, CDD and CWD are shown in Fig. 5. With regard to SDII which is defined as the ratio of annual rainfall to the number of rainy days, PERSIANN and PERSIANN-CDR show close agreement with CPC over Western US except along the west coast and over the Sierra Nevada Mountains. Over Eastern US, PERSIANN-CCS and PERSIANN-CDR underestimate SDII values. This result is of particular interest to PERSIANN-CDR because it is bias adjusted using ground observations and while it maintains the same patterns and quantities of rainfall as CPC data (See Fig. 2), SDII results indicate that while PERSIANN-CDR maintains similar accumulated rainfall depths as CPC in a monthly scale, it overestimates the number of rainy days. It should be noted that PERSIANN captures SDII values over Gulf states but depicts a spurious overestimation over central US. As for CDD, all the three products reasonably mirror the patterns observed in CPC with PERSIANN-CDR outperforming the other two products. CWD results were similar to those of CDD, with PERSIANN-



CDR outperforming PERSIANN and PERSIANN-CCS while overestimating CWD over Florida and showing a slight underestimation along the Rocky Mountains.

With respect to R10mm, PERSIANN-CDR outperforms PERSIANN and PERSIANN-CCS, mirroring the patterns observed in CPC. PERSIANN shows close performance to CPC but with underestimation. PERSIANN-CCS shows overestimation in R10mm over the Northwestern states. Results obtained for R99pTOT and R95pTOT are similar. PERSIANN shows an overestimation in the Midwestern states, consistent with SDII. Among the three products, PERSIANN-CDR outperforms the other. PERSIANN-CDR mirrors similar patterns as those of observed in CPC though with underestimation. It can be seen that PERSIANN-CDR underestimation in R10mm, R95pTOT and R99pTOT is increasing as the quantile is increased. This behaviour should be taken into consideration when using PERSIANN-CDR for extreme value analysis in engineering applications. The results of extreme indices analysis are summarized in Table 5. Previous studies show that the capabilities of satellite precipitation datasets to estimate heavy precipitation rate improves at higher temporal accumulations (Mehran & AghaKouchak, 2014).

## 4. Global comparison of PERSIANN products

Several studies have been devoted to the evaluation of the PERSIANN products against ground-based observations over different regions of the globe. Evaluation studies conducted over Iran showed that PERSIANN adequately captures the precipitation patterns of mean annual and seasonal precipitation although underestimating the amount of rainfall (Baranizadeh et al., 2012). Evaluation performed at a daily temporal scale (Katiraie-Boroujerdy et al., 2013) showed that PERSIANN and GPCP-adjusted-PERSIANN exhibit a good performance over the mountainous regions while underperforming over the coastal region of the Caspian Sea. A Recent study (Alijanian et al., 2017) which was conducted over a longer time period (2003-2012) showed that PERSIANN-CDR outperforms other satellite-based products in detecting heavy rainfall events over Iran. Similar conclusions were reported from evaluation studies performed over China which highlighted that PERSIANN-CDR accurately captures the spatial and temporal patterns of extreme rainfall, in particular, the eastern China monsoon region. However, it has been shown that PERSIANN-CDR underperforms in dry regions such as the Tibetan Plateau and the Taklamakan desert (Miao et al., 2015).

Global assessment of satellite-based products is complicated primarily due to the non-existence of ground-based measurements over oceans. Additionally, the paucity of dense rain gauge networks over land in many regions around the world is a major challenge given that precipitation variability over land is more complicated than over oceans because of topographic effects. Therefore, unlike the previously mentioned studies where a specific PERSIANN product is evaluated against ground-based observations, this section aims to provide a comparison between the three PERSIANN products, PERSIANN, PERSIANN-



CCS and PERSIANN-CDR, over the (60°S - 60°N) globe for the period (2003-2015). It should be noted that none of the products is used as a baseline for comparison but rather the performance of each product is compared against others in order to obtain insight into the properties of each product over different geographical and climatological regions.

Figure 7 illustrates that the PERSIANN products generally show similar global spatial distribution of mean annual precipitation. It can be seen that equatorial Pacific, the eastern Indian Ocean, the Amazon and western Sub-Saharan Africa are the wettest parts of the globe. However, differences are observed in the estimates of PERSIANN-CDR which are substantially lower over West Africa compared to PERSIANN and PERSIANN-CCS. Significant differences are also observed over the Amazon.

Figure 10 (a) shows the average annual precipitation for the period (2003-2015) over continents and oceans as estimated by each of the three products. This comparison is intended to provide insightful conclusions about the (Land vs Ocean) performance. As shown in Fig. 10, PERSIANN-CDR is consistently estimating higher precipitation rates over the oceans than PERSIANN-CCS and PERSIANN. On the other hand, PERSIANN-CCS estimates higher precipitation rates than PERSIANN and PERSIANN-CDR over most continents except Oceania. Additionally, it is clear that PERSIANN consistently estimates
lower precipitation rates than the other two products over both oceans and land with the exception of Africa.

Figures 8 and 9 show the mean annual zonal precipitation, i.e. across latitudes (60°S - 60°N), and the mean annual meridional precipitation, i.e. across longitudes (180°W - 180°E). While Fig. 8 may demonstrate differences in the algorithms due to climatological variations across latitudes, Fig. 9 is a mere reflection of the observations in Fig. 10 (a) as the proportion of land to ocean mass varies across longitudes. The two figures corroborate the results observed from Fig. 10. It can be seen that
PERSIANN-CDR estimates are higher over latitudes where oceans represent a larger proportion than land mass such as (60°S - 20°S) and similarly for longitudes such as (180°W - 120° W). On the other hand, PERSIANN-CCS estimates are equivalent or higher than PERSIANN-CDR across latitudes where land mass is larger than oceans such as (0 - 20°N) and similarly for longitudes such as (0 - 40°E). Moreover, it can be seen from Fig. 6 that the three products show an off-equatorial peak
approximately at 8°N which represents the mean location of the Intertropical Convergence Zone (ITCZ). Analysis of Global Precipitation Climatology Project (GPCP) zonal precipitation for the period (1979-2003) demonstrated similar results with a peak rainfall to the north of the equator approximately at 8°N (Gruber & Levizzani, 2008).

In order to provide better insight into the behaviour of the PERSIANN algorithms over land and oceans, the estimation of
30 extreme precipitation events is examined using the same group of extreme indices used for the evaluation over the CONUS (See Table 4).

As shown in Fig. 10, the three products maintain a similar SDII, with PERSIANN-CDR exhibiting generally lower values than the other two. The largest divergence between the three products appears in the Mid Pacific and Mid Indian Ocean regions. Figure 11 maps the SDII index and shows this as well. For CDD, the three products largely agree over land areas, but begin





to show differences over the oceans. They do show similar spatial patterns in terms of areas with the highest CDD values (Fig. 11), but PERSIANN tends to show more widespread areas of high CDD, followed by PERSIANN-CCS, then PERSIANN-CDR. Perhaps not surprising, the inverse behaviour is present for CWD (Fig. 10 and Fig. 11). The R10mm index shows PERSIANN-CDR with the highest values of the three products over the oceans, but lower values than the other two over land

(with the exception of Oceania and Europe). The three products generally show agreement on the extreme precipitation total, percentile-based indices (R95pTOT and R99pTOT). The largest areas of disagreement over land appear over South America and Africa. For oceans, these index values are most dissimilar between products over the Mid Pacific and Indian Ocean regions.

**5. Conclusions**

The objective of this article was to provide a summary of the differences between PERSIANN products algorithms, an evaluation of the products over the CONUS and a comparison between the products at a quasi-global scale.

The evaluation results of PERSIANN products over the CONUS indicate that the three products have generally good

correlation with Climate Prediction Center (CPC) ground-based data, and high values of Probability of Detection (POD) with averages of 0.8, 0.9 and 0.9, for PERSIANN, PERSIANN-CCS and PERSIANN-CDR respectively. The False Alarm Ration (FAR) of the three products are relatively low: 0.22 (PERSIANN), 0.29 (PERSIANN-CCS) and 0.29 (PERSIANN-CDR). PERSIANN-CDR surpasses the performance of the other two products in terms of replicating similar spatial precipitation patterns to those of CPC data. On the other hand, PERSIANN-CCS shows a slightly different pattern with overestimation in

the Northwestern states. While PERSIANN both underestimates rainfall rates in the states of Washington, Oregon and north California and overestimates over the Midwestern states. The superiority of PERSIANN-CDR is attributed to the bias adjustment of PERSIANN-CDR on a monthly scale using GPCP data. However, it is important to mention that while the results demonstrate the good performance of PERSIANN-CDR over long time scales compared to CPC, it tends to deviate (mostly underestimate) the amount of extreme rainfall on a daily scale. This emphasizes that careful attention must be paid to

problems involving extreme value analysis using PERSIANN-CDR, in particular those related to engineering applications such as infrastructure design.

Quasi-global comparison of precipitation products is important to assess the potential contrasts in products over land and ocean, and in different climatic zones. Overall, the results of global comparison indicate that the three products exhibit similar

behaviour in terms of mean annual zonal precipitation between latitudes 20°S-20°N while deviate from each other out of this equatorial belt. As for the differences between the products over land and ocean, an interesting pattern is that in the context of mean annual rainfall, PERSIANN-CDR estimates higher rainfall than the other products over the oceans while PERSIANN-



CCS estimates higher rainfall over continents except for Europe. Over both land and ocean, PERSIANN estimates of mean annual rainfall is the lowest. This comparison shall not be taken out of this context and be considered as an evaluation study because while PERSIANN and PERSIANN-CCS are purely derived from satellite observations, PERSIANN-CDR is bias adjusted using ground observations. This leads to expected deviations when comparing the products. The purpose of this

relative intercomparison was to highlight the discrepancies and differences in data products.

PERSIANN algorithms continue to improve and evolve. Recent developments including integrating deep learning approaches, adding water vapor channel information (Tao et al., 2017), using PMW data for bias adjustment of PERSIANN-CCS (Karbalaee et al., 2017), incorporating MODIS and CloudSat information (Nasrollahi et al., 2013), using probability matching methods to improve warm rainfall detection in PERSIANN-CCS. Preliminary results show that the application of deep learning

techniques in precipitation estimation is enhancing the performance of the algorithm in the detection of Rain/No Rain events. Algorithm performance is further enhanced, especially during the winter season, by utilizing the water vapor channel as an additional input to the algorithm. As for probability matching method, it is used to make bias correction of PERSIANN-CCS estimates and improve warm rainfall detections. Results of this research indicate more significant improvement in high latitudes compared to low latitudes. Although not yet operational, these methods show potential for integration in the near real-

time PRESIANN products.

*Competing interests*. The authors declare that they have no conflict of interest.

*Acknowledgement*. This research was partially supported by the ICIWaRM of the US Army corps of Engineering, UNESCO's

G-WADI program, Cooperative Institute for Climate and Satellites (CICS) program (NOAA prime award #NA14NES4320003, subaward # 2014-2913-03) for OHD-NWS student fellowship, Army Research Office (award # W911NF-11-1-0422), National Science Foundation (NSF award # 1331915), Department of Energy (DoE prime award # DE-IA0000018) and California Energy Commission (CEC Award # 300-15-005).

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





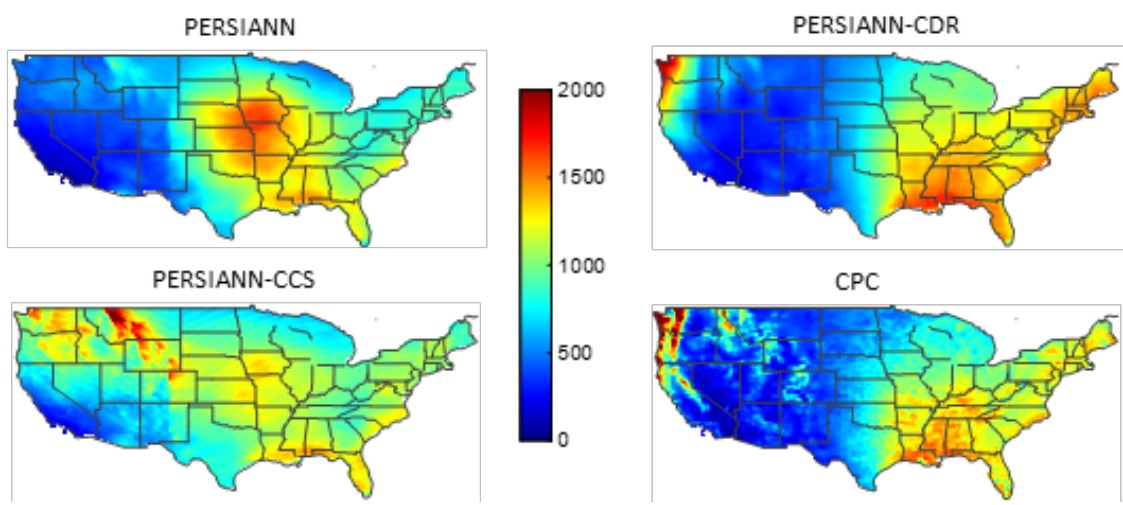

**Figure 2: Average annual precipitation (mm/year) for the period (2003-2015) over the CONUS.**



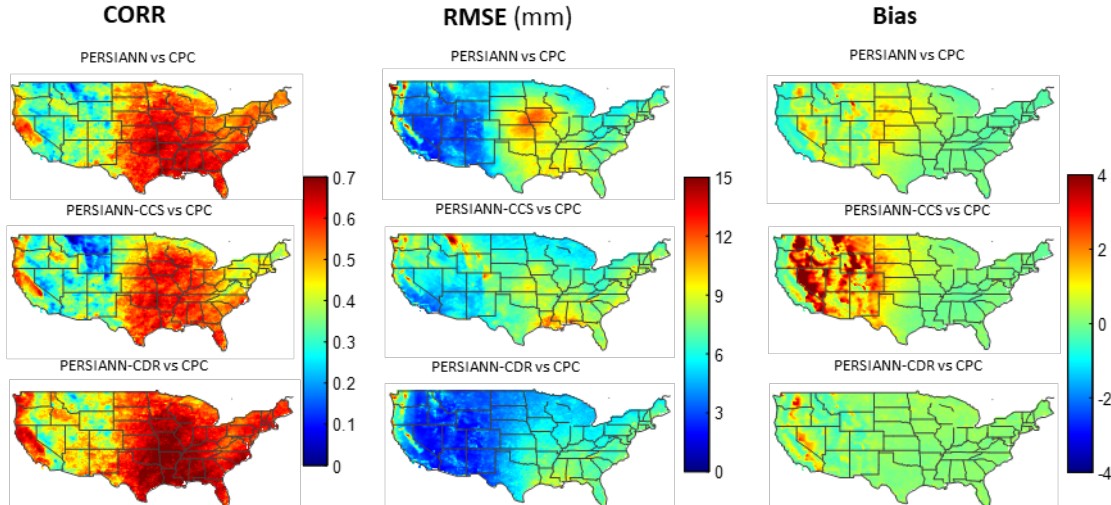

**Figure 3: Correlation (CORR), Root Mean Square Error (RMSE) and Bias of PERSIANN, PERSIANN-CCS and PERSIANN-CDR against CPC for the period (2003-2015) over the CONUS.**





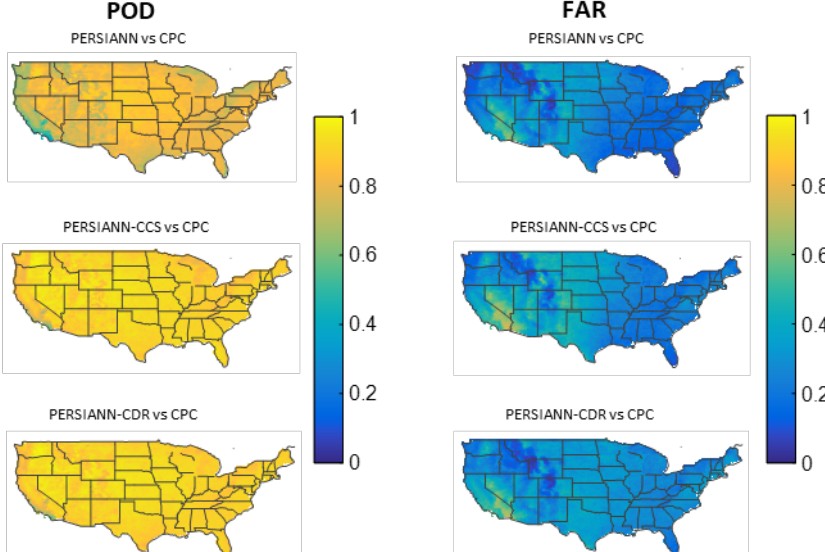

**Figure 4: Probability of Detection (POD) and False Alarm Ratio (FAR) of PERSIANN, PERSIANN-CCS and PERSIANN-CDR against CPC for the period (2003-2015) over the CONUS.**





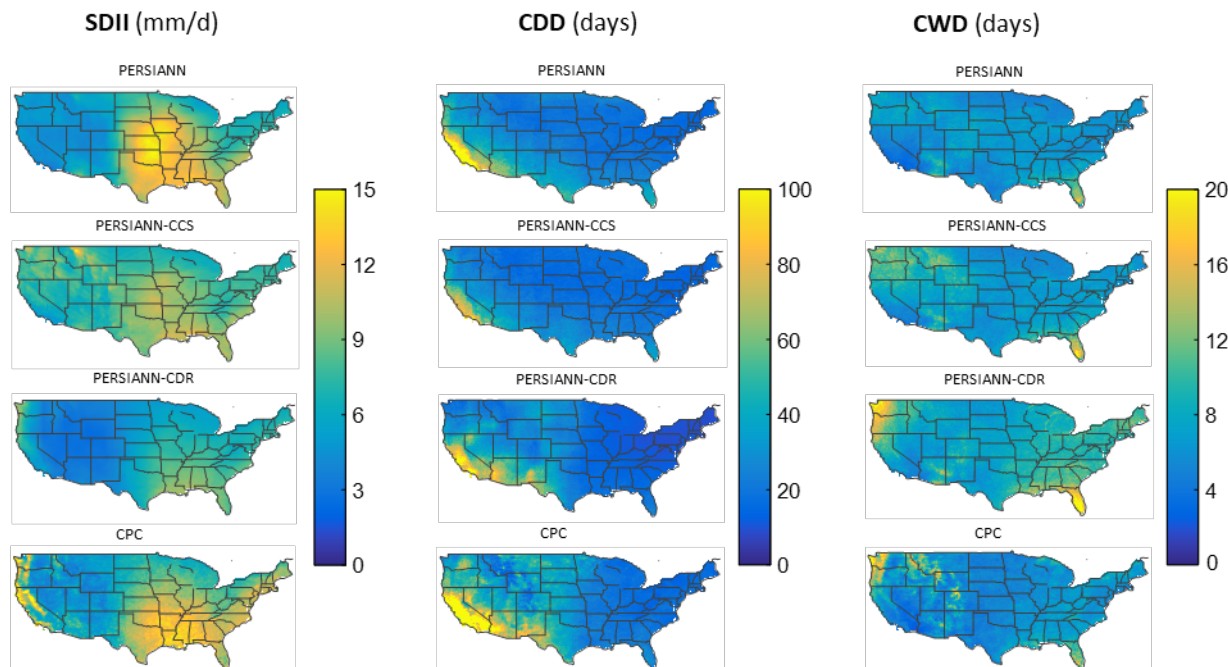

**Figure 5: Extreme Precipitation Indices (SDII, CDD and CWD) for PERSIANN, PERSIANN-CCS and PERSIANN-CDR over the CONUS (See Table 4 for definition of the indices).**





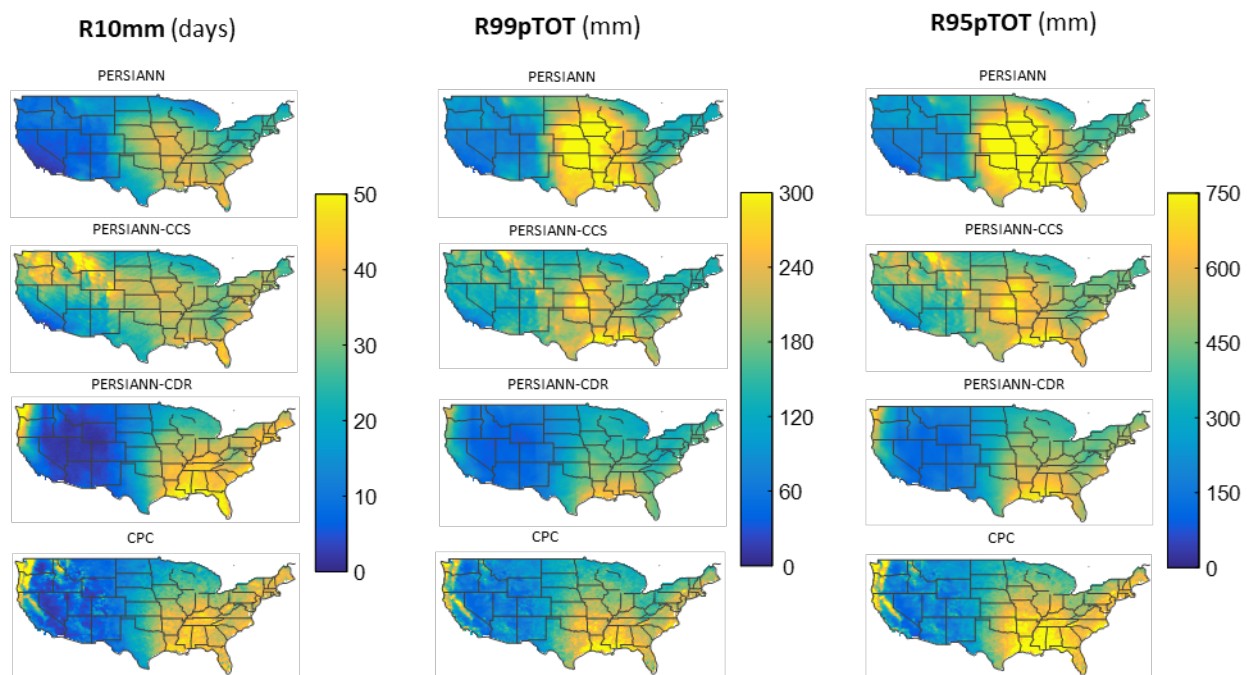

**Figure 6: Extreme Precipitation Indices (R10mm, R99pTOT and R95pTOT) for PERSIANN, PERSIANN-CCS and PERSIANN-CDR over the CONUS (See Table 4 for definition of the indices).**



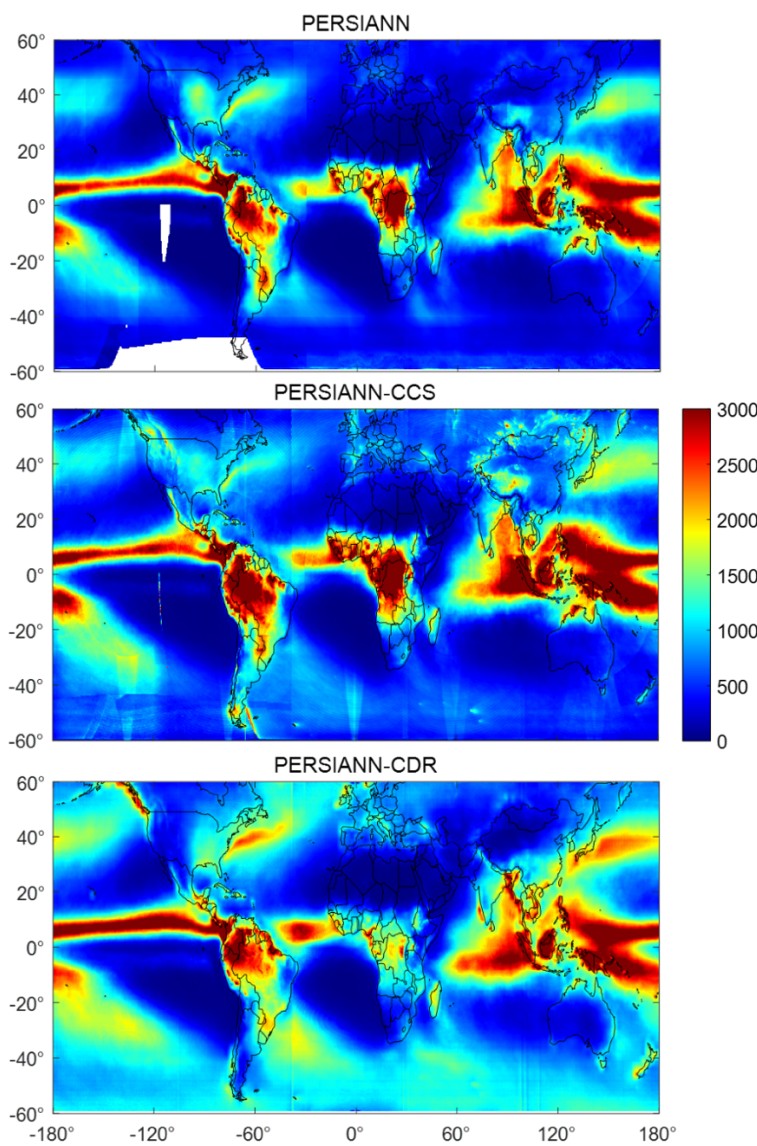

**Figure 7: Mean annual precipitation (mm) for PERSIANN, PERSIANN-CCS and PERSIANN-CDR.**





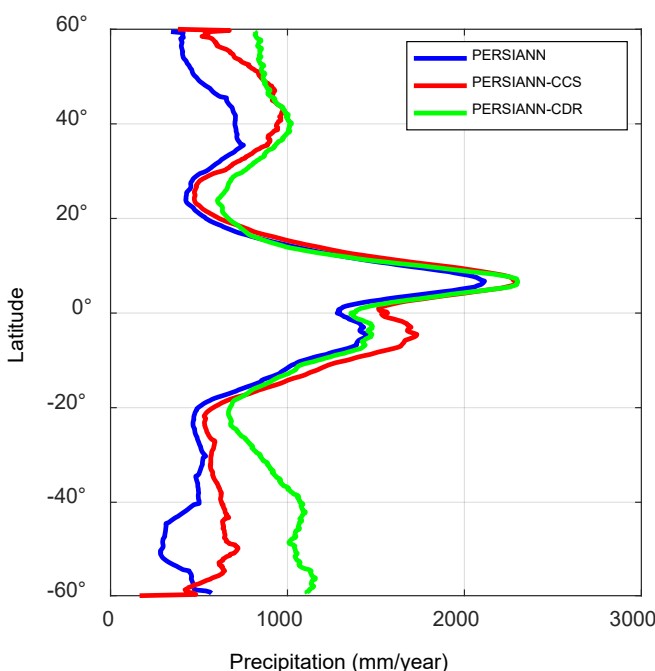

Figure 8: Mean annual zonal precipitation (mm/year) for PERSIANN, PERSIANN-CCS and PERSIANN-CDR.





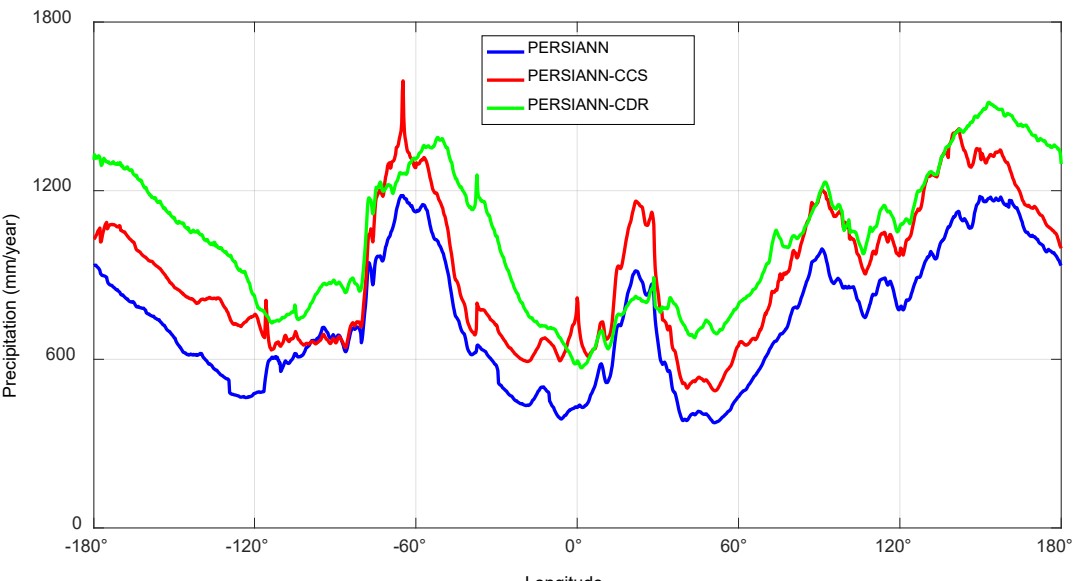

**Figure 9: Mean annual meridional precipitation (mm/year) for PERSIANN, PERSIANN-CCS and PERSIANN-CDR.**



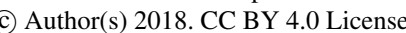



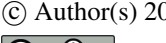



**Figure 10: Mean annual precipitation and Extreme Indices (SDII, CDD, CWD, R10mm, R99pTOT and R95pTOT) over continents and oceans for PERSIANN, PERSIANN-CCS and PERSIANN-CDR.**





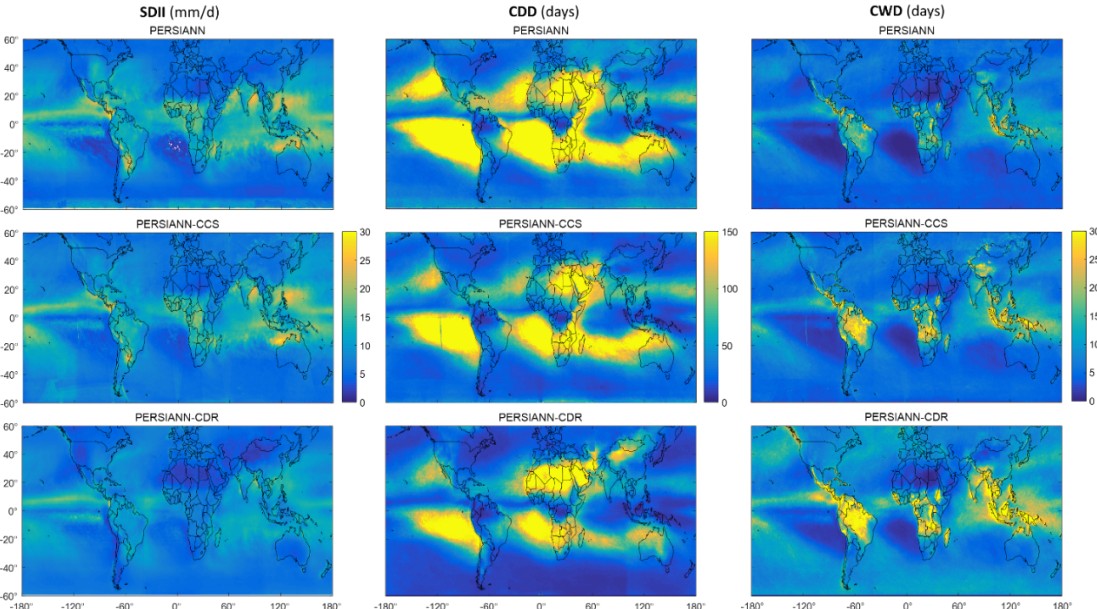

**Figure 11: Extreme Precipitation Indices (R10mm, R99pTOT and R95pTOT) for PERSIANN, PERSIANN-CCS and PERSIANN-CDR over the globe. (See Table 4 for definition of the indices).**




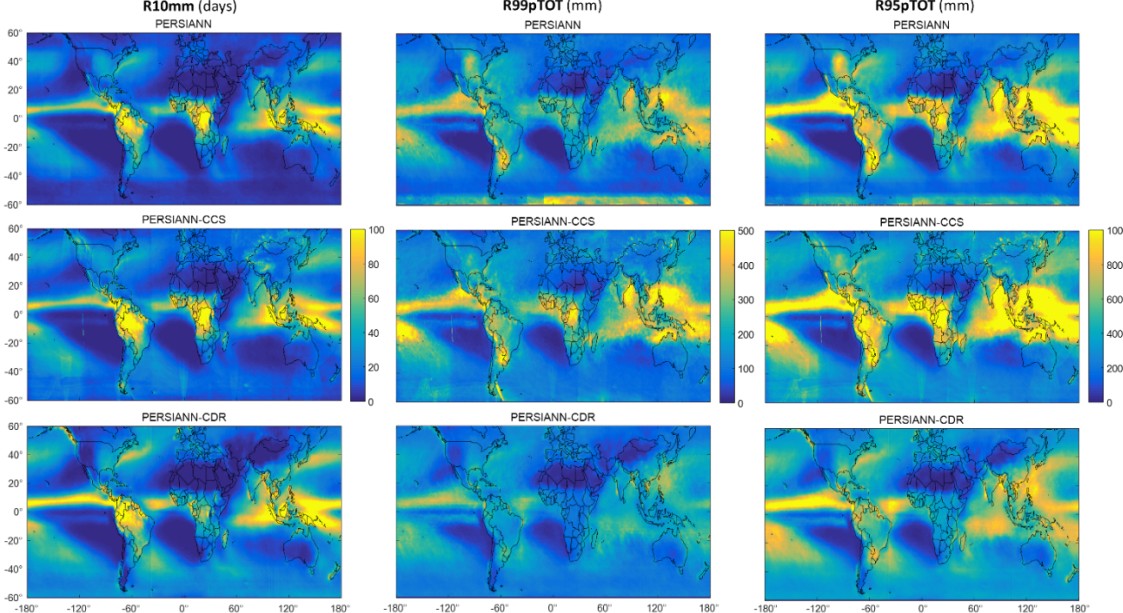

**Figure 12: Extreme Precipitation Indices (R10mm, R99pTOT and R95pTOT) for PERSIANN, PERSIANN-CCS and PERSIANN-CDR over the globe. (See Table 4 for definition of the indices).**



**Table 1: Basic attributes of the PERSIANN, PERSIANN-CCS and PERSIANN-CDR products.**

| Product | Availability Period | Spatial Coverage | Temporal Resolution | Spatial Resolution | Time Delay |
|---|---|---|---|---|---|
| PERSIANN | Mar 2000 - Present | 60°S - 60°N | 1 hour | 0.25° x 0.25° | ~2 days |
| PERSIANN-CCS | Jan 2003 - Present | 60°S - 60°N | 1 hour | 0.04° x 0.04° | ~1 hour |
| PERSIANN-CDR | Jan 1983 - Present | 60°S - 60°N | 1 day | 0.25° x 0.25° | ~3months |





**Table 2: Algorithm attributes of PERSIANN, PERSIANN-CCS and PERSIANN-CDR products**

|  | PERSIANN | PERSIANN-CCS | PERSIANN-CDR |
|---|---|---|---|
| Primary Input data | GEO longwave infrared images (10.2-11.2 μm) | GEO longwave infrared images (10.2-11.2 μm) | ISCCP-B1 GEO satellite infrared gridded data |
| Data for training of the model (parameter estimation) | LEO Passive microwave information | LEO Passive microwave information | NCEP Stage IV precipitation data (0.04° x 0.04°) |
| Use of Passive Microwave (PMW) Data | Yes | No | No |
| Batch mode (fixed parameters) vs Recursive mode (non-fixed parameters) | Recursive Mode (non-fixed parameters) | Batch Mode (fixed parameters) | Batch Mode (fixed parameters) |
| Bias Correction | No | No | Yes |
| Data used for Bias Correction | - | - | GPCP monthly precipitation data (2.5° x 2.5°) |



5    **Table 3: Summary of comparison metrics for the PERSIANN family of products against CPC for the period (2003 to 2015) over the CONUS.**

| Metrics | PERSIANN | PERSIANN-CCS | PERSIANN-CDR |
|---|---|---|---|
| CORR | 0.48 | 0.43 | 0.55 |
| RMSE (mm) | 6.50 | 7.08 | 5.19 |
| Bias | 0.10 | 0.72 | 0.14 |
| POD | 0.80 | 0.90 | 0.90 |
| FAR | 0.22 | 0.29 | 0.29 |



**Table 4: Definition of extreme precipitation indices used in the analysis.**

| Index | Definition | Unit |
|-------|-----------|------|
| R99pTOT | Annual total precipitation when daily precipitation amount on a wet day > 99 percentile | mm |
| R95pTOT | Annual total precipitation when daily precipitation amount on a wet day > 95 percentile | mm |
| SDII | Simple daily intensity index | mm day$^{-1}$ |
| R10mm | Annual count of days when daily precipitation amount ≥10 mm | Days |
| CWD | Annual maximum number of consecutive days with daily precipitation amount ≥1 mm | Days |
| CDD | Annual maximum number of consecutive days with daily precipitation amount < 1 mm | days |



**Table 5: Statistics of the PERSIANN family of products for the period (2003 to 2015) over the CONUS.**

| Index | PERSIANN | PERSIANN-CCS | PERSIANN-CDR | CPC |
|---|---|---|---|---|
| Annual Precipitation (mm/year) | 793.41 | 978.51 | 916.45 | 852.32 |
| SDII (mm/day) | 7.58 | 7.98 | 5.80 | 8.41 |
| CDD (days) | 27.44 | 22.23 | 24.22 | 34.19 |
| CWD (days) | 6.36 | 7.38 | 9.27 | 6.72 |
| R10mm (days) | 20.94 | 30.42 | 24.77 | 25.77 |
| R99pTOT (mm) | 164.32 | 166.34 | 120.48 | 158.51 |
| R95pTOT (mm) | 415.03 | 445.70 | 340.11 | 422.79 |