# Peer review of "The PERSIANN Family of Global Satellite Precipitation Data: A Review and Evaluation of Products"

_Hydrology and Earth System Sciences, 2018_

## Referee Comment (RC1) · Anonymous Referee #1 · 11 May 2018

The manuscript "The PERSIANN Family of Global Satellite Precipitation Data: A Review and Evaluation of Products" by Nguyen Phu et al. meets the scope of the journal and is thus worth considering for publication pending some modifications.

The paper is needed because the PERSIANN family of satellite-derived precipitation products has considerably increased since the first product was made available some time ago. The end users thus experience a bit of difficulty in using the right product for their own purpose. Therefore, the need for a paper that tells the whole story with some clear numbers.

Here are my comments on the present version of the manuscript:

1) A more thorough discussion on the differences stemming from the analysis of the

three products both over CONUS and over the globe seems necessary to me. For example, the relatively large differences found between PERSIANN/PERSIANN-CCS and PERSIANN-CDR are not discussed enough in my view. The authors briefly mention that this is due to the satellite-only character of the two products. However, I think a more thorough discussion would be useful for the reader.

2) Given the focus of the journal, also a clearer mention to the potential of the three products for hydrological and Earth science applications would seem appropriate. The authors would help in this way the potential user in choosing the correct products for his/her need.

3) The paper is generally written in an acceptable English. However, it needs an accurate proofreading to eliminate language deficiencies that prevent a smooth reading and fast understanding of the concepts. The reviewer started correcting, but soon found out that the errors were far too many. Nothing dramatic, but it is annoying. A fine combing is necessary and the authors ought to do it.

---

## Referee Comment (RC2) · Anonymous Referee #2 · 22 May 2018

General Comments

The paper addresses a satellite-based precipitation product called PERSIANN. This is one of many journal articles devoted to this product but the authors justify the need for this one on the basis of recent updates and upgrades. In fact, the article discusses three PERSIANN products: PERSIANN, PERSIANN-CCS, and PERSIANN-CDR. I think the authors should improve their description of the justification for the three families of products. Clearly, this could be confusing for users who would prefer to have one product for all their needs. Lower resolution products, if needed, could always be available by the upscaling of the higher resolution products. Perhaps a simple schematic with a time line could provide an easy to understand justification.

Specific Comments
A concern is the use of the CPC data set for the product evaluation. The authors should comment on the uncertainty of the product. Are there gridded uncertainty maps associated with the CPC product? If not, what are the obstacles to producing them? Any comparison with a ground-based reference is incomplete without characterizing the uncertainty of the reference. Also, just a cautionary note that the correlation coefficient for skewed random variables (like rainfall) tends to be overestimated. Is the bias additive or multiplicative?

Overall, it is disappointing to me that space-based products have little skill unless corrected with simple, old rain gauges. Not the authors' fault but something worth commenting on.

The paper says little, if anything, regarding hydrologic applications of the product. The journal is about hydrology, after all. . .. Is the skill adequate for hydrologic applications? Which applications? Should we be impressed with the skill? I'd like to see authors' perspective on the question. The authors warn against using the product for engineering design, and that's good but in many parts of the world this might still be the best option available.

The authors should improve the quality of the figures. Figure 1 is practically useless. Other figures showing the US are too small and not properly aligned. The continuous color scale obstructs the spatial features. Perhaps 6-8 color categories would show them better.

The entire paper should be carefully edited and use active voice throughout the paper.

---

## Referee Comment (RC3) · Anonymous Referee #3 · 23 May 2018

General comments This short paper presents three PERSIANN satellite-based precipitation products. A comparison of the products with the CPC ground-based precipitation is performed over the United States from 2003 to 2015, as well as an intercomparison between products at the global scale. While this broad overview may be valuable to the research community and the topic fits the scope of the journal, there are some questions to address.

1. Applications of these precipitation products, especially for hydrological applications should be more discussed in the perspective of the presented performances. For example there is no discussion in the manuscript on the impact of uncertainty from PERSIANN-CCS on the GPM IMERG product.

2. The interpretation of the comparison results needs to be expanded throughout the

manuscript. More information is needed regarding satellite precipitation uncertainty structure. For example how do you explain PERSIANN-CCS climatological features in Fig. 2? Only gauge correction in PERSIANN-CDR seems to correct efficiently the PERSIANN and PERSIANN-CCS climatologies. How can this be explained? A discussion of precipitation products assumptions, strengths, and limitations should be added in the context of this evaluation. Aspects like remote sensing physics, precipitation physics and algorithmic influence should be addressed. For example regarding PERSIANN-CCS: under the assumption relating colder Tbs to higher rain rates using PDF matching, the resulting precipitation estimates could be influenced by the climatology of (cold) Tbs generated by specific types of precipitation systems, e.g. mesoscale convective systems in the Great Plains.

3. Can the authors elaborate on the representativeness of the CPC comparison analysis outside the U.S. (regarding all products), and especially at locations devoid of gauge networks (regarding PERSIANN-CDR)?

4. It is not fair to compare a gauge-adjusted product (PERSIANN-CDR) with satellite-only precipitation products (PERSIANN and PERSIANN-CCS). Besides it is important to use an independent reference for an objective comparison and evaluation. Finally the ground reference should present consistent accuracy across CONUS, which may not be the case with CPC if the gauge network density is not homogeneous.

5. The evaluation is performed at the daily time scale at the finest. As precipitation varies across space and time scales, the concluding remarks should recall this comparison scale. An evaluation at the native resolution of the products (i.e. hourly for PERSIANN and PERSIANN-CCS) would be more insightful and relevant. Can the authors comment on the representativeness of their findings and their dependence on resolution?

Specific comments:

1. p.3 l. 15-20: what about NOAA precipitation products?

2. p.6 l.16: "it combines all ground-based information sources": does it combine also radar data?

3. What is the precipitation rate threshold used in categorical indices like POD and FAR?

4. p.7 ll.17-20: why not using the volumetric indices?

---

## Author Comment (AC1) · 3 Jul 2018

Reviewer #1: The manuscript "The PERSIANN Family of Global Satellite Precipitation Data: A Review and Evaluation of Products" by Nguyen Phu et al. meets the scope of the journal and is thus worth considering for publication pending some modifications. The paper is needed because the PERSIANN family of satellite-derived precipitation products has considerably increased since the first product was made available some time ago. The end users thus experience a bit of difficulty in using the right product for their own purpose. Therefore, the need for a paper that tells the whole story with some clear numbers. Here are my comments on the present version of the manuscript:

[Figure]

We greatly appreciate your comments and suggestions which we believe will result in a much-improved version of the manuscript. The manuscript has been revised according to some of your comments while incorporation of other comments is in progress.

1) A more thorough discussion on the differences stemming from the analysis of the three products both over CONUS and over the globe seems necessary to me. For example, the relatively large differences found between PERSIANN/PERSIANN-CCS and PERSIANN-CDR are not discussed enough in my view. The authors briefly mention that this is due to the satellite-only character of the two products. However, I think a more thorough discussion would be useful for the reader.

Response: We agree that a more thorough discussion of the analysis results is necessary. The manuscript is currently being revised to extend the discussion and interpretation of analysis results.

2) Given the focus of the journal, also a clearer mention to the potential of the three products for hydrological and Earth science applications would seem appropriate. The authors would help in this way the potential user in choosing the correct products for his/her need.

Response: We thank you for this constructive comment. In the new version, the manuscript includes a discussion about the suitability of each PERSIANN product to different hydrological and water resources management applications taking into consideration their characteristics and the analysis results.

3) The paper is generally written in an acceptable English. However, it needs an accurate proofreading to eliminate language deficiencies that prevent a smooth reading and fast understanding of the concepts. The reviewer started correcting, but soon found out that the errors were far too many. Nothing dramatic, but it is annoying. A fine combing is necessary and the authors ought to do it.

Response: Thank you for pointing out to this limitation. These language deficiencies
will be fixed in the revised version.

---

## Author Comment (AC2) · 3 Jul 2018

Reviewer #2:

We greatly appreciate your comments and suggestions which we believe will result in a much-improved version of the manuscript. The manuscript has been revised according to some of your comments while incorporation of other comments is in progress.

General Comments

I think the authors should improve their description of the justification for the three families of products. Clearly, this could be confusing for users who would prefer to have one product for all their needs. Lower resolution products, if needed, could always be available by the upscaling of the higher resolution products. Perhaps a simple

schematic with a time line could provide an easy to understand justification.

Response: Thank you for this suggestion. In the revised version, text has been added in section 2 to illustrate the purpose of each product. The main justification for the existence of three products is that they provide data at different lag times and different time coverage. Therefore, they are tailored for different hydrometeorological applications.

Specific Comments

A concern is the use of the CPC data set for the product evaluation. The authors should comment on the uncertainty of the product. Are there gridded uncertainty maps associated with the CPC product? If not, what are the obstacles to producing them? Any comparison with a ground-based reference is incomplete without characterizing the uncertainty of the reference. Also, just a cautionary note that the correlation coefficient for skewed random variables (like rainfall) tends to be overestimated. Is the bias additive or multiplicative?

Response: Based on your suggestion, the revised manuscript provides additional information about CPC data such as the number and average density of gauges involved in the interpolated product, the interpolation method . . .etc. Also, references have been cited for additional information. The bias is additive; however, it is relative bias (not absolute). This has been illustrated in the revised manuscript in figures captions.

Overall, it is disappointing to me that space-based products have little skill unless corrected with simple, old rain gauges. Not the authors' fault but something worth commenting on.

Response: The use of IR imagery (cloud top temperature) as a proxy in precipitation estimation requires bias correction with ground observations. However, as estimation of precipitation from satellites continues to evolve by including other information (e.g. water vapor channel) and developing new algorithms, we anticipate that satellite-based precipitation will attain higher accuracy.

The paper says little, if anything, regarding hydrologic applications of the product. The journal is about hydrology, after all.... Is the skill adequate for hydrologic applications? Which applications? Should we be impressed with the skill? I'd like to see authors' perspective on the question. The authors warn against using the product for engineering design, and that's good but in many parts of the world this might still be the best option available.

Response: We thank you for this constructive comment. In the new version, the manuscript includes a discussion about the suitability of each PERSIANN product to different hydrological and water resources management applications taking into consideration their characteristics and the analysis results.

The authors should improve the quality of the figures. Figure 1 is practically useless. Other figures showing the US are too small and not properly aligned. The continuous color scale obstructs the spatial features. Perhaps 6-8 color categories would show them better.

Response: Based on your suggestion, in the revised manuscript, most figures have been replotted for better quality.

The entire paper should be carefully edited and use active voice throughout the paper.

Response: Thank you for pointing out to this limitation. The language deficiencies will be fixed in the revised version.

---

## Author Comment (AC3) · 3 Jul 2018

**Reviewer 3**

General comments

This short paper presents three PERSIANN satellite-based precipitation products. A comparison of the products with the CPC ground-based precipitation is performed over the United States from 2003 to 2015, as well as an intercomparison between products at the global scale. While this broad overview may be valuable to the research community and the topic fits the scope of the journal, there are some questions to address.

We greatly appreciate your comments and suggestions which we believe will result in a much-improved version of the manuscript. The manuscript has been revised according

to some of your comments while incorporation of other comments is in progress.

1. Applications of these precipitation products, especially for hydrological applications should be more discussed in the perspective of the presented performances. For example there is no discussion in the manuscript on the impact of uncertainty from PERSIANN-CCS on the GPM IMERG product.

Response: We thank you for this constructive comment. In the new version, the manuscript includes a discussion about the suitability of each PERSIANN product to different hydrological and water resources management applications taking into consideration their characteristics and the analysis results.

2. The interpretation of the comparison results needs to be expanded throughout the manuscript. More information is needed regarding satellite precipitation uncertainty structure. For example how do you explain PERSIANN-CCS climatological features in Fig. 2? Only gauge correction in PERSIANN-CDR seems to correct efïñĄciently the PERSIANN and PERSIANN-CCS climatologies. How can this be explained? A discussion of precipitation products assumptions, strengths, and limitations should be added in the context of this evaluation. Aspects like remote sensing physics, precipitation physics and algorithmic inïñĆuence should be addressed. For example regarding PERSIANN-CCS: under the assumption relating colder Tbs to higher rain rates using PDF matching, the resulting precipitation estimates could be inïñĆuenced by the climatology of (cold) Tbs generated by speciïñĄc types of precipitation systems, e.g. mesoscale convective systems in the Great Plains.

Response: We agree that a more extensive interpretation of the analysis results is necessary. Currently, the manuscript is being revised to extend the discussion about comparison results both over CONUS and globally.

3. Can the authors elaborate on the representativeness of the CPC comparison analysis outside the U.S. (regarding all products), and especially at locations devoid of gauge networks (regarding PERSIANN-CDR)?

[Figure]

Response: Thank you for this comment. Due to the wide variability in climate and precipitation regimes across the CONUS, we expect that the performance patterns observed for each product might be representative of regions outside the CONUS with similar climate and precipitation regimes. However, we refrain from making strong conclusions about this as other unknown factors might have an important role. Furthermore, it should be noted that previous studies have investigated the performance of the different PERSIANN prodiucts outside the US at local spatial scales (i.e. countries or catchments); some of these studies are referred to in this manuscript.

4. It is not fair to compare a gauge-adjusted product (PERSIANN-CDR) with satellite only precipitation products (PERSIANN and PERSIANN-CCS). Besides it is important to use an independent reference for an objective comparison and evaluation. Finally the ground reference should present consistent accuracy across CONUS, which may not be the case with CPC if the gauge network density is not homogeneous.

Response: It is true that PERSIANN-CDR is inherently different than PERSIANN and PERSIANN-CCS since it incorporates ground-based information due to bias adjustment. However, the aim of the global inter-comparison is to reveal general patterns about the products behavior in different geographical regions. It is not our intention to perform any kind of evaluation since neither of the products can be considered as a baseline. As for the evaluation of each product over the CONUS, it is true that PERSIANN-CDR is not completely independent of CPC because of bias adjustment. However, it should be noted that the bias adjustment of PERSIANN-CDR is on a monthly scale, meanwhile, the evaluation is performed at a finer resolution of daily scale.

5. The evaluation is performed at the daily time scale at the finest. As precipitation varies across space and time scales, the concluding remarks should recall this comparison scale. An evaluation at the native resolution of the products (i.e. hourly for PERSIANN and PERSIANN-CCS) would be more insightful and relevant. Can the authors comment on the representativeness of their findings and their dependence

on resolution?

Response: We agree that the results should be interpreted in light of the evaluation temporal scale (daily). In the revised manuscript, additional discussion will be added to reflect on this. Hourly evaluation for the products is not feasible due to unavailability of high spatial resolution CPC data at an hourly temporal resolution. The hourly CPC data is available up to 2002 and available at a coarse resolution of 2 x 2.5 degrees. Another alternative for a reference data is ST4 data which doesn't have good accuracy over the western US. Furthermore, It worth mentioning that both PERSIANN and PERSIANN-CCS have been evaluated at an hourly scale at some locations over the CONUS (Hong et al., 2004).

Specific comments

1. p.3 l. 15-20: what about NOAA precipitation products?

Response: CPC CMORPH is the NOAA satellite-based precipitation product. In the revised manuscript, "NOAA" has been added to clearly illustrate this.

22. p.6 l.16: "it combines all ground-based information sources": does it combine also radar data?

Response: Thank you for pointing out to this. The sentence has been clarified in the revised manuscript, as CPC data include all gauge-based information available at the Climate Prediction Center (CPC) but does not include radar data. See https://www.esrl.noaa.gov/psd/data/gridded/data.unified.daily.conus.html.

3. What is the precipitation rate threshold used in categorical indices like POD and FAR? 4. p.7 ll.17-20: why not using the volumetric indices?

Response: The threshold for POD and FAR calculation is 0.1 mm. We opted not to use the most commonly categorical indices of POD and FAR due to its widespread use in literature instead of the volumetric indices.
References Hong, Y., Hsu, K., Sorooshian, S. and Gao, X. (2004). Precipitation estimation from remotely sensed imagery using an artificial neural network cloud classification system. J. Appl. Meteor., 43, 1834-1852.

---

## Author Response (AR1)

**Reviewer #1:**

The manuscript "The PERSIANN Family of Global Satellite Precipitation Data: A Review and Evaluation of Products" by Nguyen Phu et al. meets the scope of the journal and is thus worth considering for publication pending some modifications. The paper is needed because the PERSIANN family of satellite-derived precipitation products has considerably increased since the first product was made available some time ago. The end users thus experience a bit of difficulty in using the right product for their own purpose. Therefore, the need for a paper that tells the whole story with some clear numbers. Here are my comments on the present version of the manuscript:

We greatly appreciate your comments and suggestions which we believe resulted in a much-improved version of the manuscript. The manuscript has been revised according to your comments.

1)   A more thorough discussion on the differences stemming from the analysis of the three products both over CONUS and over the globe seems necessary to me. For example, the relatively large differences found between PERSIANN/PERSIANN-CCS and PERSIANN-CDR are not discussed enough in my view. The authors briefly mention that this is due to the satellite-only character of the two products. However, I think a more thorough discussion would be useful for the reader.

**Response:** Thank you for your comment. Discussion about the differences between products has been extended. See lines (28-31, page 6), (22-28, page 7) in section 3 as well as the extended discussion in the conclusion section.

2) Given the focus of the journal, also a clearer mention to the potential of the three products for hydrological and Earth science applications would seem appropriate. The authors would help in this way the potential user in choosing the correct products for his/her need.

**Response:** Thank you. Based on your suggestion, text has been added to illustrate the suitability of each products for different hydrometeorological applications based on their characteristics (see lines 15-22, page 4). In addition, the potential of each product for different hydrologic applications in light of the analysis results has been discussed in the conclusion. See lines (18-32, page 10) and lines (1-17, page 11).

3) The paper is generally written in an acceptable English. However, it needs an accurate proofreading to eliminate language deficiencies that prevent a smooth reading and fast understanding of the concepts. The reviewer started correcting, but soon found out that the errors were far too many. Nothing dramatic, but it is annoying. A fine combing is necessary and the authors ought to do it.

**Response:** Thank you for pointing this out. The manuscript has been carefully edited to remove language deficiencies and grammatical errors.

**Reviewer #2:**

We greatly appreciate your comments and suggestions which we believe resulted in a much-improved version of the manuscript. The manuscript has been revised in accordance you're your comments and suggestions.

**General Comments**

I think the authors should improve their description of the justification for the three families of products. Clearly, this could be confusing for users who would prefer to have one product for all their needs. Lower resolution products, if needed, could always be available by the upscaling of the higher resolution products. Perhaps a simple schematic with a time line could provide an easy to understand justification.

**Response:** Thank you for your comment. In the revised manuscript, text has been added in section 2 to illustrate the purpose of each product (see lines 15-22, page 4). The main justification for the existence of three products is that they provide data at different lag times and different time coverage. Thus, they are tailored to different hydrometeorological applications.

**Specific Comments**

A concern is the use of the CPC data set for the product evaluation. The authors should comment on the uncertainty of the product. Are there gridded uncertainty maps associated with the CPC product? If not, what are the obstacles to producing them? Any comparison with a ground-based reference is incomplete without characterizing the uncertainty of the reference. Also, just a cautionary note that the correlation coefficient for skewed random variables (like rainfall) tends to be overestimated. Is the bias additive or multiplicative?

**Response:** Based on your suggestion, the revised manuscript provides additional information about CPC data such as the number and average density of gauges involved in the interpolated product, the interpolation method …etc. Also, references have been cited for additional information. (See lines 14-20, page 6).

The bias is additive; however, it is relative bias (not absolute). This has been illustrated in the revised manuscript in figures captions.

Overall, it is disappointing to me that space-based products have little skill unless corrected with simple, old rain gauges. Not the authors' fault but something worth commenting on.

**Response:** The use of IR imagery (cloud top temperature) as a proxy in precipitation estimation requires bias correction with ground observations. However, as estimation of precipitation from satellites continues to evolve by including other information (e.g. water vapor channel) and developing new algorithms, we anticipate that satellite-based precipitation will attain higher accuracy.

The paper says little, if anything, regarding hydrologic applications of the product. The journal is about hydrology, after all.... Is the skill adequate for hydrologic applications? Which applications? Should we be impressed with the skill? I'd like to see authors' perspective on the question. The authors warn against using the product for engineering design, and that's good but in many parts of the world this might still be the best option available.

**Response:** We thank you for this constructive comment. In the new version, the manuscript includes a discussion about the suitability of each PERSIANN product to different hydrological and water resources management applications taking into consideration their characteristics and the analysis results. See lines (15-22, page 4), (18-32, page 10) and (1-17, page 11).

The authors should improve the quality of the figures. Figure 1 is practically useless. Other figures showing the US are too small and not properly aligned. The continuous color scale obstructs the spatial features. Perhaps 6-8 color categories would show them better.

**Response:** Based on your suggestion, most figures in the revised manuscript have been replotted with a better quality. Also, a categorical color scale is used in several figures.

The entire paper should be carefully edited and use active voice throughout the paper.

**Response:** Thank you for pointing this out. The manuscript has been carefully edited to remove language deficiencies and grammatical errors.

**Reviewer 3**

**General comments**

This short paper presents three PERSIANN satellite-based precipitation products. A comparison of the products with the CPC ground-based precipitation is performed over the United States from 2003 to 2015, as well as an intercomparison between products at the global scale. While this broad overview may be valuable to the research community and the topic fits the scope of the journal, there are some questions to address.

We greatly appreciate your comments and suggestions which we believe resulted in a much-improved version of the manuscript. The manuscript has been revised in accordance you're your comments and suggestions.

1. Applications of these precipitation products, especially for hydrological applications should be more discussed in the perspective of the presented performances. For example, there is no discussion in the manuscript on the impact of uncertainty from PERSIANN-CCS on the GPM IMERG product.

**Response:** Thank you. Based on your suggestion, text has been added in section 5 to illustrate the potential of each product for different hydrologic applications considering the analysis results. See lines (18-32, page 10) and lines (1-17, page 11).

2. The interpretation of the comparison results needs to be expanded throughout the manuscript. More information is needed regarding satellite precipitation uncertainty structure. For example how do you explain PERSIANN-CCS climatological features in Fig. 2? Only gauge correction in PERSIANN-CDR seems to correct efficiently the PERSIANN and PERSIANN-CCS climatologies. How can this be explained? A discussion of precipitation products assumptions, strengths, and limitations should be added in the context of this evaluation. Aspects like remote sensing physics, precipitation physics and algorithmic influence should be addressed. For example regarding PERSIANN-CCS: under the assumption relating colder Tbs to higher rain rates using PDF matching, the resulting precipitation estimates could be influenced by the climatology of (cold) Tbs generated by specific types of precipitation systems, e.g. mesoscale convective systems in the Great Plains.

**Response:** Thank you for your comment. Discussion about the differences between products has been extended. See lines (28-31, page 6), (22-28, page 7) in section 3 as well as the extended discussion in the conclusion section. It should also be noted that several results in the article has not been fully interpreted either because further research is needed to attribute the sources of discrepancies or due to the lack of ground observations. The latter case is specific for the global inter-comparison.

3. Can the authors elaborate on the representativeness of the CPC comparison analysis outside the U.S. (regarding all products), and especially at locations devoid of gauge networks (regarding PERSIANN-CDR)?

**Response:** Thank you for this comment. Due to the wide variability in climate and precipitation regimes across the CONUS, we expect that the performance patterns observed for each product might be representative of regions outside the CONUS with similar climate and precipitation regimes. However, we refrain from making strong conclusions about this as other unknown factor might have an important role. Furthermore, it should be noted that previous studies have investigated the performance of the different PERSIANN products outside the US at local spatial scales (i.e. countries or catchments); some of these studies are referred to in this manuscript.

4. It is not fair to compare a gauge-adjusted product (PERSIANN-CDR) with satellite only precipitation products (PERSIANN and PERSIANN-CCS). Besides it is important to use an independent reference for an objective comparison and evaluation. Finally, the ground reference should present consistent accuracy across CONUS, which may not be the case with CPC if the gauge network density is not homogeneous.

**Response:** It is true that PERSIANN-CDR is inherently different than PERSIANN and PERSIANN-CCS since it incorporates ground-based information due to bias adjustment. However, the aim of the global inter-comparison is to reveal general patterns about the products behavior in different geographical regions. It is not our intention to perform any kind of evaluation since neither of the products can be considered as a baseline.

As for the evaluation of each product over the CONUS, it is true that PERSIANN-CDR is not completely independent of CPC because of bias adjustment. However, it should be noted that the bias adjustment of PERSIANN-CDR is on a monthly scale, meanwhile, the evaluation is performed at a finer resolution of daily scale. We acknowledge the limitations of this evaluation, namely the non-uniform density of gauges in CPC product and the fact that PERSIANN-CDR is adjusted on a monthly scale using CPC. However, CPC remains the most accurate product to be used for evaluation.

5. The evaluation is performed at the daily time scale at the finest. As precipitation varies across space and time scales, the concluding remarks should recall this comparison scale. An evaluation at the native resolution of the products (i.e. hourly for PERSIANN and PERSIANN-CCS) would be more insightful and relevant. Can the authors comment on the representativeness of their findings and their dependence on resolution?

**Response:** We agree that the results should be interpreted in light of the evaluation temporal scale (daily). The revised manuscript recalls this fact wherever needed. It should be noted that hourly evaluation for the products is not feasible due to unavailability of high spatial resolution CPC data at an hourly temporal resolution.

**Specific comments**

1. p.3 l. 15-20: what about NOAA precipitation products?

**Response:** CPC CMORPH is the NOAA satellite-based precipitation product. In the revised manuscript, "NOAA" has been added to clearly illustrate this.

22. p.6 l.16: "it combines all ground-based information sources": does it combine also radar data?

**Response:** Thank you for pointing this out. The sentence has been clarified in the revised manuscript, as CPC data include all gauge-based information available at the Climate Prediction Center (CPC) but does not include radar data.

See https://www.esrl.noaa.gov/psd/data/gridded/data.unified.daily.conus.html.

3. What is the precipitation rate threshold used in categorical indices like POD and FAR? 4. p.7 ll.17-20: why not using the volumetric indices?

**Response:** The threshold for POD and FAR calculation is 0.1mm. We opted to use the most commonly categorical indices of POD and FAR due to its widespread use in literature instead of the volumetric indices.

---

## Author Response (AR2)

**Editor Decision**: Publish subject to minor revisions (review by editor) (23 Oct 2018) by Remko Uijlenhoet

Comments to the Author:

Dear authors,

Your revised manuscript has now been reviewed by the same three reviewers as the original paper. All reviewers agree that the manuscript has significantly improved with respect to the previous version. I tend to agree with that. The remaining issues are mainly of an editorial nature. I invite you to consider the final remarks and suggestions made by two of the reviewers. In particular, I would like to ask you to:

- Pay explicit attention in the abstract, discussion and conclusions to the limits of the insights provided by this study, related to the fact that the presented comparisons are performed on a daily basis. It would be great if you could speculate a bit on the implications of the presented results for the quality of the various rainfall products at the sub-daily (down to hourly) scale, which is a scale of great hydrological significance (relevant for the HESS readership).

- Take a careful look at the figures: provide the highest quality figures possible (now some of the figures seem to be somewhat vague); avoid using colors if they do not add information for the readers; and make sure that the figure legends and captions provide complete information on what each figure tries to convey.

- Finally, perform a final round of detailed spelling and grammar checks, to avoid having to correct many minor errors during the proofreading stage.

Thank you very much in advance for taking these issues into account. I will evaluate the way in which they have been dealt with myself. The updated version of the manuscript, which I look forward to receive, will not be sent out for review again.

Best regards,

Remko Uijlenhoet

We would like to thank Professor Uijlenhoet for your instructive editorship in our manuscript. We have revised the manuscript in accordance with the detailed comments provided by the reviewers.

**Reviewer #1:**

Accepted as is.

We would like to thank reviewer 1 for his/her acceptance.

**Reviewer #2:**

The authors addressed the comments from the previous review.

The limits of the insights provided by this study should be better highlighted in the abstract and in the conclusion. The goal of this paper is to provide an overview of the PERSIANN algorithms and their difference, and perform an intercomparison to highlight their strengths and limitations. The intercomparison exercise at the daily time scale provides limited insight into the strengths and limitations of the PERSIANN and PERSIANN-CCS products for applications at their native resolution (hourly). PERSIANN and PERSIANN-CCS have certainly been designed for applications at sub-daily scale. Applications mentioned in the manuscript require sub-daily precipitation, e.g. development of Intensity-Duration-Frequency curves or hydrologic applications such as floods or flash-floods prediction. PERSIANN-CCS is used in the GPM IMERG product at 30-min / 0.1 degree resolution. It would be relevant and interesting to have a discussion a this scale of application as mentioned in the previous review comments. A detailed discussion regarding the satellite precipitation uncertainty structure involving algorithm assumptions, remote sensing physics, and precipitation physics is probably beyond the scope of this overview. It is however recommended to highlight clearly that this discussion on the PERSIANN products is representative at the daily scale and cannot be readily extrapolated to other and specifically shorter time scales.

It is also recommended to proofread the manuscript for typos.

We would like to thank the reviewer for his/her constructive comments and suggestions. We have revised the abstract and manuscript as suggested by the reviewer and highlighted the limitations of this study in the abstract and conclusion.

We had a native English speaker, Mr. Dan Braithwaite, co-author of this manuscript go through to check typos and grammar errors.

**Reviewer #3:**

The revised version of the paper is much improved and I recommend its publication. I'm still not impressed with the quality of the figures but the publisher should make the final decision on whether they are adequate. Figure 1 is still unnecessary, there is practically one color and the figure could be easily summarized in one or two sentences. In Figure 10, colors and symbols? Why both?

We would like to thank this reviewer for his/her advice on the figure quality. We decided to remove Figure 1 as suggested by this reviewer since it is not very informative. We replotted Figure 9 (Figure 10 previously) in black and white since this figure does not need to be in colors as suggested by reviewer 3.

[revised manuscript text omitted]